# Social Learning: Towards Collaborative Learning with Large Language Models

## Abstract

We introduce the framework of "social learning" in the context of large language models (LLMs), whereby models share knowledge with each other in a privacy-aware manner using natural language. We present and evaluate two approaches for knowledge transfer between LLMs. In the first scenario, we allow the model to generate abstract prompts aiming to teach the task. In our second approach, models transfer knowledge by generating synthetic examples. We evaluate these methods across diverse datasets and quantify memorization as a proxy for privacy loss. These techniques inspired by social learning yield promising results with low memorization of the original data. In particular, we show that performance using these methods is comparable to results with the use of original labels and prompts. Our work demonstrates the viability of social learning for LLMs, establishes baseline approaches and highlights several unexplored areas for future work.

## 1 Introduction

Increasingly, large language models are considered a crucial building block for agents that can reason [27], use tools [21] and adapt to environmental cues [20, 42] for many real-world tasks. As such, personal assistants are now commonly powered by such models [29] while larger entities, e.g. companies, can also have their own agents. When considering networks of personal agents, the ability to transfer information and foster collaboration is highly desirable. For instance, a spam detector can be collaboratively maintained by sharing newly detected spam templates.

Collaboration among language models to solve complex problems involves various research areas [37], for example task planning [15], information retrieval [11, 43] and information exchange [19]. LLMs have shown impressive capabilities at performing novel tasks by following natural language instructions or using a limited number of examples [5, 38]. This suggests that natural language might become a viable means of knowledge transfer for personal agents. However, a critical concern is how to ensure the privacy of users is upheld by preventing the leakage of sensitive information between agents.

In this work, we introduce the paradigm of privacy-aware "social learning" to transfer knowledge between LLMs. We take inspiration from the theory of social learning as defined by Bandura & Walters [4] which proposes that new behaviors can be acquired by observing and imitating others. Indeed, mechanisms of social learning have proven highly effective in persistent multi-agent systems by allowing agents to benefit from the accumulated learning of others [2, 25]. The resulting framework enables agents to generate examples and instructions tailored for task-specific information transfer with an emphasis on safeguarding the privacy of shared examples and knowledge. We posit that this framework is advantageous as it provides knowledge transference between models in a human-interpretable way without sharing private data.

The key contributions of our work are (1) proposing and formalizing the concept of social learning for LLM-driven agents; (2) suggesting baseline implementations of social learning and benchmarking them across a diverse set of tasks and (3) establishing metrics to measure private data leakage, and using them to demonstrate the benefits of social learning whilst preserving privacy.

## 2 Problem Setting & Methods

Language models have made significant strides in generating effective responses based on instructions, spanning domains like planning and memory [37]. However, the inclusion of private data brings forth

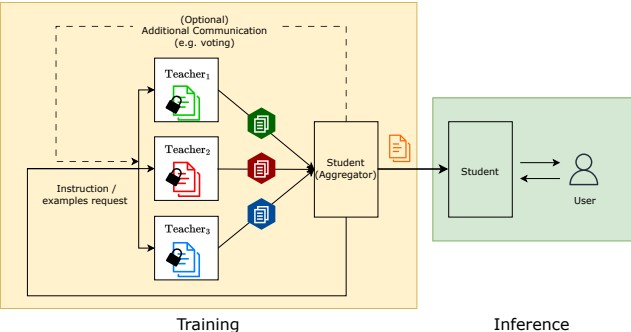

Figure 1: An illustration of our social learning framework. Teachers have access to private data that they cannot directly share. The student does not have access to such data. Instead it relies on the teachers to create instructions or non-private examples to teach it the task. After receiving these instructions, the student aggregates them into a single prompt. This prompt is used by the student at inference time to respond to a user's queries.

new challenges, including navigating data ownership, preserving privacy, and securely transferring knowledge. In this work, we introduce the social learning framework as a tailored response to these challenges. Specifically, we explore an environment where information about a task is communicated from multiple teachers to a student through text-based interactions, within predefined constraints aimed at preserving the privacy of original examples.

As a real-world example of such an environment, consider the task of detecting whether a message received through Short Message Service (SMS) is spam or not. Let us assume that we have asked $m$ users to act as annotators and classify their messages as spam or not spam. The goal is to use this data to enable a new user's phone to automatically detect whether a new incoming message is spam or not. However, while users may agree to perform the annotation, they seldom want to share the contents of their messages due to privacy concerns. Therefore the goal is to send informative messages based on labeled data available locally on each user's phone without communicating the contents of any message.

## 2.1 SOCIAL LEARNING PROTOCOL

We provide a canonical definition of social learning in this section by considering $m$ agents $\mathcal{T}_1, \ldots, \mathcal{T}_m$, called teachers that teach a task (e.g. yes/no question answering) to another agent $\mathcal{S}$, called the student. Each teacher has access to its own silo of data $\mathcal{D}_{\mathcal{T}_i}$ which contains a distinct subset of examples for the task. Meanwhile, the student does not have access to any training data. A user queries the student at inference time to solve new, unseen instances of the task. As such, the goal is to transfer the knowledge of the teachers to the student so that it can successfully respond to a query.

Similar to standard machine learning models, we consider two operation modes for this environment: training and inference. During training the agents collaborate without any input to transfer task-related knowledge whereas at inference time the student relies on this transferred knowledge to answer the specific instance of the task. Therefore, the student can augment its knowledge (stored in $\mathcal{D}_{\mathcal{S}}$) by communicating with teachers during training and subsequently relies on the accumulated knowledge to answer queries at inference time.

At training time, part of the role of the student is that of an **aggregator** where it must select a subset of the information provided to it by the teachers. In this work, we only consider the most basic version of the student at inference which replies to a user input by appending the input to a prompt, querying its language model, and returning the continuation. The whole process is illustrated in Figure 1.

A solution to the problem of how to teach the student can be to send all the data accessible by the teachers to the student and have it concatenate all of these data points to create the final prompt. In this case, the student receives all the knowledge and the task is reduced to generating a good response based on the available data. However, it is important to consider cases where this is not possible, for example because of privacy constraints. In particular, we consider the scenario where

the original examples accessible by the teachers contains private data that should not be shared with other parties. Therefore, the goal is to teach the student without sharing such private information which automatically excludes the possibility of sharing the original examples. Similarly, we focus on the cases where the user's query to the student contains private data and therefore can not be directly shared with the teachers. In our evaluations, we consider directly sharing the original examples of the teachers as a baseline to compare our methods against.

## 2.2 METHODS

The mechanisms of knowledge transfer in our work are inspired by social learning theory [4]. The theory outlines models of observational learning amongst humans and we use two of these as the basic models of communication in our framework. While a combination of these basic models is most likely more effective, in this work we only look at the performance when they are employed separately. These models are simple enough that they satisfy additional constraints which allows us to avoid the need for abilities in language models that are yet to be perfected. We refer the interested reader to Appendix A where we provide an overview of these constraints and their motivations.

### 2.2.1 VERBAL SOCIAL LEARNING: SHARING INSTRUCTIONS

In the **verbal instruction** model from social learning theory, a behavior is described in detail and a participant is instructed in how to engage in the behavior.

Conversely, LLMs are able to perform new tasks based on short, textual instructions describing the tasks in question [24]. Previous work has also shown that these instructions can be generated by prompting an instruction-tuned LLM with examples and then asking it to complete the instruction for them [14].

Similar to the verbal instruction model, we can thus ask teacher agents to generate instructions based on their silo of private data. These instructions are then shared with the student who integrates the instructions in its prompt. In this work, when using this model, we focus on the scenario where there is only a single teacher. We apply this simplification to avoid the need for an aggregation mechanism that merges multiple instructions and leave developing such mechanisms for future work.

### 2.2.2 LIVE MODELS: SHARING EXAMPLES

In the **live models** method from social learning theory, an individual demonstrates the desired behavior and the learner imitates.

Conversely, a technique used that allows LLMs to perform well on a new task is including examples of that task in the prompt [5], a technique called few-shot learning. Even including a few examples can greatly improve the downstream performance.

One option for teaching using this learning model is sharing examples from the teacher's private dataset. However, this method compromises privacy which is why we only consider it as a baseline. Instead, we consider sharing artificial examples that are generated based on the real data.

To let teachers generate artificial examples, we make use of their language models. In particular, given the capability of language models to follow the format of the input and replicate it [34], the continuation of a few-shot prompt can be expected to contain new examples. Hence, to generate a new artificial example, each teacher selects $n_{gen}$ examples from its private set and generates artificial examples by providing them as the few-shot prompt to its language model, using the model to generate a continuation without any additional instructions.

The continuations are generated by querying the model with temperature sampling with temperature $\tau$ and selecting the top scoring (based on perplexity) $k$ continuations. Some of these continuations might be discarded due to concerns such as privacy or faulty generation while the rest are sent to the aggregator, the component responsible for generating the final prompt for the student. The aggregator then picks from the at most $n_{gen} \cdot k$ generated examples, and adds the selected ones to the student's prompt.

# 3 RELATED WORK

## 3.1 LLMS AND AGENTS

Zero-shot or few-shot prompting has been shown to be highly effective for transfer learning, notably by Brown et al. [5]. In such approaches, a large pre-trained language model is zero-shot or few-shot prompted by being shown examples of the desired behaviour, without training, to perform a new task. Variations on these methods such as chain-of-thought prompting [39] have shown that even simple prompt modifications can have a substantial impact on target task performance [39, 7] and enable new capabilities.

There is a large pre-existing body of work focused on multi-agent based communication via dialogue to solve complex tasks [10, 32]. The motivation is that by cross-agent interaction, LLMs can collectively exhibit enhanced performance by aggregating their strengths. Multiple works have focused on debate between LLMs to improve output of models. For instance, [12] allow multiple language model instances to propose and debate their responses and reasoning processes. Their findings indicate that this approach significantly enhances mathematical and strategic reasoning across a number of tasks. Perez et al. [28] also propose a debate procedure to verify the accuracy and safety of generated content. However, in these scenarios, the concept of agents having access to separate datasets is not considered.

Most similar to our work, Zeng et al. [44] introduce a modular framework that allows multimodal models to exchange information with each other and capture new capabilities using zero-shot transfer. Their approach does not require fine tuning and aims to capitalize on the different types of knowledge contained by models capturing different modalities.

## 3.2 FEDERATED LEARNING

Federated learning [18, 22, 17] is a technique for training models on decentralized data without collecting any of this data in a central place. Instead, a central server coordinates the fleet of participants during the training process. In each round of training, a subset of participants is sampled. Each participant receives the current weights of the model, uses their local data to update them, and then sends back the gradients. The server combines all the model updates across participants and uses them to update the model of the next iteration.

Social learning is similar to federated learning in that no raw data is meant to be transmitted and that the participants aim to jointly learn to perform a task. However, in contrast to federated learning, social learning does not update any model weights and instead works solely by exchanging information expressed in natural language. This has a few advantages:

1. All components are agnostic to the specific models used. Teachers and students can be based on different model sizes, architectures and weights. All they need to be able to do is to input and output natural language.

2. Text is more compact than gradients. In federated learning today, it would be prohibitively expensive to send full updates for the largest foundation models. With social learning, everything is expressed in text fitting a prompt, which can easily be transmitted across networks.

3. Text is much more interpretable than gradients. One can read what teachers produce and analyze it.

While social learning is distinctively different from federated learning, some of its concepts can be transferred across to the social learning setting. In our privacy analysis in Section 5 for example, we adapt Secret Sharer [6], a technique that is also popular in federated learning.

# 4 EXPERIMENTS

In order to assess the effectiveness of the methods we discussed in Section 2.2, we evaluate their performance on different tasks in this section. Since the challenges involved in social learning are new, it also requires its own task suite. In this work, we propose a set of tasks with different properties and challenges and use them for benchmarking. We provide an overview of the benchmarking suite and the properties of each task in Appendix B. In most of the experiments, we use instances of PaLM 2 [3] models, specifically PaLM 2-S, to power both the teachers and the student. Since we need

```
The following examples are privately shared with you and will not be
given to the participants. Describe the format (any special markings used
), and general patterns and any other useful generic notes that you can
find based on these examples. What you write will be the only hint given
to the participant and they are expected to output correct replies in the
 right format.
<Original Examples>

Task format with detailed instructions:
```

Figure 2: The prompt used to generate instructions for a task.

| $n$ | Type | Lambada | BoolQ | GSM8K | SMSSpam | SMS Spam (With Class) | Random Insertion |
|---|---|---|---|---|---|---|---|
| 0 | - | 69.8 | 68.1 | $0.0^{\alpha}$ | 14.2 | 92.7 | 22.0 |
| 1 | Original | **86.7** | 89.8 | **63.6** | 59.1 | 94.3 | 55.6 |
| | Generated | **86.7** | 70.5* | **63.9** | 90.2* | 92.6* | 53.6* |
| 2 | Original | **87.3** | 90.1 | 64.2 | 77.2 | 94.9 | 70.0 |
| | Generated | 86.7 | 88.6* | 63.2* | 88.2* | 92.2* | 65.9* |
| 4 | Original | 87.6 | 90.4 | **63.6** | **86.8** | 95.4 | **69.8** |
| | Generated | 88.0 | 85.6* | **63.6** | **87.8** | 90.2* | **69.7** |
| 8 | Original | 88.4 | 90.5 | **64.1** | 96.0 | 96.8 | 74.5 |
| | Generated | 88.1 | 88.7* | **63.4** | 86.5* | 91.5* | 69.2* |
| 16 | Original | 88.4 | **90.4** | **63.6** | 96.5 | 97.0 | **73.5** |
| | Generated | 89.0 | **90.0** | **63.7** | 88.0* | 91.1* | **72.4** |

Table 1: Performance of PaLM 2-S with different methods on different datasets. A star marks a statistically significant difference between performance using original and generated examples. We bold cells where no statistically significant difference was detected to emphasize that in many cases the examples generated using social learning perform as well as the original ones. The average accuracy across 5 runs is reported. Table 7 reports the same values with more precision. A description of datasets and the formatting used for providing them to the models is given in Appendix B.

$^{\alpha}$GSM8K uses a special format to mark the answer. The model inevitably always fails when no instruction or examples are provided to clarify this special format. Adding the prefix stated in Figure 5 in the Appendix to clarify the format yields an accuracy of 16.38%.

the model to follow instructions when doing instruction generation, to ease comparison, we use the instruction-tuned version of the models in all of our experiments.

To account for the randomness arising from temperature sampling and the distribution of the dataset between teachers, we repeat each experiment 5 times and report the mean. We also perform significance testing, as described in Appendix C. This lets us systematically evaluate whether there are meaningful differences between using original data and synthetic data generated through social learning.

## 4.1 LIVE MODELS: SHARING EXAMPLES

We follow the process outlined in Section 2.2.2 with $m = 8$ teachers and compare the performance of a prompt with $n$ generated examples for different values of $n$ against several baselines. The dataset is distributed between teachers randomly so all teachers will have the same data distribution. The zero-shot performance of the model on the task institutes a low bar baseline. As a high bar, we consider the performance of doing few-shot learning with $n$ private examples from one of the teachers, equivalent to asking that teacher to directly solve the task. Note that this is not feasible in practice and thus is a high bar since sending private examples of a teacher, or querying one teacher with inputs given to the student violates their privacy. Therefore, we do not aim to outperform this baseline but to show that we can perform comparably using the generated examples.

In most of our experiments we use a basic aggregation mechanism where the aggregator picks one of the artificially generated candidates at random. We call this aggregator the random aggregator.

We start by considering the scenario where the student's language model is the same as the teachers'. Since the only difference between the teachers and the student in this case is the set of examples they can access, we can compare the effect of using generated examples instead of real ones more clearly. The results are shown in Table 1 and highlight various patterns that give insight on effectiveness of generating artificial examples. We now discuss several of these patterns in detail.

For the majority of tasks, we observe no significant difference between using original private examples and the generated ones, especially when the number of examples is high enough, e.g. $n = 16$. This is especially interesting since we observe that these generated examples are sufficiently different from the real ones. We confirm this in Appendix D where we report a high average normalized distance between each generated example and the prompt used for generating it. We note that this investigation is different from measuring the amount of data leakage which we investigate in Section 5 as the examples can be different and yet still contain sensitive information.

The main exception where a difference can be observed between generated and real examples is the spam detection task. Based on our observations, we conjecture that one of the underlying reasons that makes generating artificial examples for this task more challenging is that the language model favors not spam examples over spam examples. Boolean Questions is another task where the model struggles when given generated examples, though the gap closes when the number of examples is large enough. In this task we also observe that the language model seems to strongly favor questions with a yes answer, suggesting that the favor of one class is a re-occurring challenge in generating examples for classification tasks. For Boolean Questions we also observe another challenge that the language model tends to generate questions that do not have a yes or no answer. We provide some qualitative examples of both good and bad generations in Appendix J.

Finally, we observe that generating factual examples is not essential for transferring knowledge. For example, we observe that some of the generated examples and provided solutions in the GSM8K task can be wrong without hurting performance. As shown in prior work [23], the demonstrations are not only useful to show the mapping between the input and the label but are also important to clarify the format and the input and label distributions. We conjecture that in these cases the model mainly relies on its own intrinsic ability to map the input to the label while using the demonstrations to learn the other aforementioned aspects of the task. We highlight that these aspects are sometimes essential to a good performance on the task. Indeed, on the GSM8K task, thinking step by step is part of the format learned from the examples which significantly improves performance [39].

### 4.1.1 EXTENSIONS TO SHARING EXAMPLES

We additionally investigated two extensions to the above setup which we only briefly describe here with details described in the appendix.

**Teaching to a larger student**   This ability is natural to social learning since teachers only share text, enabling knowledge to be transferred between different models of different sizes and architectures. On the other hand, typical gradient-based federated learning methods such as FedAvg [22] and FedOpt [33] require the same model size and architecture to be used everywhere. Given that the largest of language models currently can be only executed on data centers, it would be especially useful to be able to transfer knowledge back to such models. In our experiments, we find this to be generally feasible in social learning, with a small drop in performance compared to teachers and student being of the same size, as is expected to be in this more difficult setting. Details and results of this setup is provided in Appendix G.

**Voting aggregator**   As an example of a more sophisticated aggregator, we evaluated an aggregator where teachers vote on their preferred examples. To be able to do this, teachers keep a hold-out dataset that is used during the voting process. After teachers generated examples using their training dataset, the aggregator sends back all received examples to the teachers to let them vote. The most popular examples are then used by the student during evaluation. We find this protocol to improve results for intermediate values of $n$, the total number of examples picked by the aggregator. We refer interested readers to Appendix H for more details and results.

### 4.2 VERBAL SOCIAL LEARNING: SHARING INSTRUCTIONS

As discussed in Section 2.2.1, sharing an instruction for the task is another possible method for social learning where the teachers are asked to generate an instruction that describes the task. In this work,

| Method | Lambada | BoolQ | GSM8K | SMSSpam | SMS Spam (With Class) | Random Insertion |
|---|---|---|---|---|---|---|
| Zero-Shot | 69.8 | 68.1 | 0.0 | 14.2 | 92.7 | 22.0 |
| Manual | 77.5 | 90.2 | 15.6 | 94.0 | 94.2 | 34.9 |
| 8-shot Original Examples | **88.4** | **90.5** | **64.1** | **96.0** | **96.8** | **74.5** |
| 8-shot PaLM 2-S Generated Examples | **88.1** | 88.7 | **63.4** | 86.5 | 91.5 | **69.2** |
| GPT3.5 Generated Instruction | 82.8 | **90.1** | 4.1 | 85.4 | **95.4** | 59.2 |
| PaLM 2-S Generated Instruction | 85.1 | 88.7 | 0.0 | **92.9** | 93.4 | 40.4 |

Table 2: Performance of PaLM 2-S when transferring knowledge using generated instructions. For each dataset, we bold the best-performing baseline and social learning method. In most cases, the generated instruction improves over directly prompting the model with the task (zero-shot). We can observe that for some of the tasks such as Lambada and Random Insertion, using generated examples performs better than using generated instructions whereas the situation is reversed for the spam detection task. The average accuracy across 5 runs is reported. Table 8 reports the same values with more precision.

we only consider the single teacher case to avoid the need for merging multiple instructions. The teacher is queried a single time to generate an instruction based on 8 examples, pointing out any patterns or special format instructions that it can observe (see the exact prompt in Figure 2). The generated instruction is directly used as the prompt for the student. As such, the aggregator in this case simply forwards the instruction.

We present the results in Table 2 for two teacher models: PaLM 2-S and OpenAI GPT3.5-Turbo. The table also includes the results for multiple baselines. In particular, we compare with the empty prompt (zero-shot) performance as the low bar to showcase the improvement observed from having an instruction. Since the instruction is generated using 8 examples, we also compare with the 8-shot performance (without instruction) using the original, private examples directly as the high bar. Finally, as an alternative, we also report results on a prompt that we wrote manually for each task. These prompts are listed in Table 6. While writing a manual prompt is not a controlled process, we report the results here to provide an approximate of what can be achieved without using social learning and simply relying on the intuition of the model developer. To simulate the prompt developers' limited access to a task's examples, the prompts were only tested and tuned with at most 2 examples from each task.

With the exception of the GSM8K task and the spam detection task with list of classes provided, we observe an accuracy that is significantly improved in comparison with zero-shot performance. The most challenging dataset for generating instruction seems to be GSM8K. We observed that the main challenge for this task is providing the instruction for the special format of the output which involves outputting the final answer after four hash (#) signs. In many of the runs, the models ignore this special format and do not include it in the instruction which leads to a zero accuracy performance. Moreover, even in some of the runs where GPT3.5 generates an instruction which includes the description of the format, the performance is usually below the manual instruction performance and much lower than sharing original or generated examples. We note that our results are based on a basic method for generating the instruction. Indeed, recent work suggests that the instruction can be significantly improved using more sophisticated generation methods. For example Yang et al. [41] report results comparable to the performance we observe with original examples by using a feedback loop in the generation process. We leave exploration of different methods to improve the instruction as future work. Interestingly, we can observe that in some tasks, namely Lambada and Random Insertion, generated artificial examples perform better than generated instructions whereas in other tasks such as spam detection, generated instruction obtains a higher accuracy. Still, in all tasks the performance is lower than the high bar of 8-shot original examples, suggesting a capacity for improvement.

## 5 MEMORIZATION

In the previous sections of the paper, we discussed how well teachers can teach students in social learning in terms of model quality. In this section, we investigate whether the instructions and examples transferred to students indeed help reduce private data leakage or not. To this end, we propose and evaluate metrics to measure how much social learning can memorize sensitive information included in the private examples.

As a first step, we first investigate how often teachers copy over one of their private examples verbatim. This can happen when the teacher repeats one of the examples given in its prompts. On all datasets we found this to be the case in fewer than 0.1% of cases, meaning the exact data point is rarely leaked. As shown in Table 5 in the Appendix, the Levenshtein distances between original and generated examples are also generally high. However, that does not necessarily mean that no sensitive parts of the original example are memorized, either verbatim or in more subtle ways.

To investigate this further, we adapt the existing Secret Sharer [6] technique for social learning. Secret Sharer is an established technique for measuring how much a given training process leads a model to memorize some of its training data. It has been used in federated learning [36, 13], making it an interesting technique to adapt to social learning.

Secret Sharer works by inserting artificial secret data points, called *canaries*, into the training data set. Injection of canaries provides access to a known set of secrets that should not be shared, making it measurable how much the secrets present in the data are memorized. To implement this, one canary is randomly sampled from a list containing $N_{SS}$ potential canaries, while the other $N_{SS} - 1$ data points that were not sampled serve as comparison elements. In our experiments, we generate canaries containing secret codes and names. This is done by using random four-digit numbers for the codes and by taking names from a dataset of the most common names given to newborns in the US in 2020 [16]. The codes or names are inserted into patterns shown in Table 13 in the appendix.

After performing training using the data containing the canary, the score assigned by the model to the canary included in the data is compared with the scores of the comparison data points that were not included in the training data. This metric, called *rank*, counts the number of comparison examples that get assigned a higher score than the canary that was actually trained on. Secret Sharer assumes a scoring function based on the model that assigns a higher score to examples that the model memorized. Since the rank is a random variable, the average of the rank across $T_{SS}$ runs is computed and used for making deductions. For example, if the model has not memorized the canary, the rank's distribution would be uniform, leading to an average rank of $\frac{N_{SS}}{2}$. In the case of perfect memorization, the rank would be 0.

To illustrate the method further, consider the example of adding the canary `The secret code is 1234` to the training set. After training, we can check how high the model's score is in that particular example as opposed to the same string with different codes. A model that only learned a high-level pattern, would not assign a significantly higher score to the string containing the particular code it was trained on whereas a model that memorized the concrete data point would.

In standard gradient descent training, the model's loss for the example can be used as the score. In social learning, we do not optimize any numerical loss and do not update any weights. Instead, the social learning process produces a string in the form of new examples or an instruction which can be added to the model's prompt. Therefore, we use the following mechanism to compute the score: Given the final prompt from social learning process and a canary, the likelihood of that canary as a continuation of the learned prompt is determined by the model. This value is normalized by the number of tokens in the canary to make it comparable to the score of other canaries. This normalized value is used as the score. We call this scoring function the *example reconstruction likelihood*. An example of this can be seen in Figure 3.

Putting everything together, a Secret Sharer experiment in social learning then works as follows:

1. A canary element and $N_{SS} - 1$ comparison elements are sampled.

2. The canary element is inserted into the training dataset of all teachers.

3. The social learning process is executed, which results in examples or an instruction generated by the teacher.

4. The example reconstruction likelihood is computed on **(a)** The canary element used in training. and **(b)** The $N_{SS} - 1$ comparison elements not used in training.

5. The rank is computed by counting how many of the comparison elements have a higher likelihood than the one we trained on.

6. The above process is repeated $T_{SS}$ times and the average rank is returned.

$P_{\text{Example3}}\Big($ *Example1: Ali goes to ____ -> school* / *Example2: Hamid is a friend of ____ -> Ali* / *Example3: Hamid goes to ____ -> school* $\Big)$

Figure 3: Example reconstruction likelihood is the score the model assigns to an original example (in blue, representing either the canary or a comparison element) which follows the generated examples. The score is only computed on the original example.

| Canary | Lambada | GSM8K |
|--------|---------|-------|
| Codes  | 435     | 467   |
| Names  | 463     | 459   |

Table 3: The average rank across 100 Secret Sharer experiments.

| Canary | Lambada | GSM8K |
|--------|---------|-------|
| Codes  | 8       | 3     |
| Names  | 7       | 4     |

Figure 4: How often rank 0 occurs across 100 Secret Sharer experiments. In a random, uniform distribution, we would expect it to occur once.

Since each experiment requires performing many social learning experiments to compute a stable average rank, running this method is costly. Therefore, we only evaluate it on two of the tasks, namely Lambada and GSM8K. Furthermore, we focus on measuring the memorization for two different types of secrets, namely numbers (as secret codes) and names, in the canary elements.

We compare the rank of an included canary with 999 other not included canaries, i.e. $N_{\text{SS}} = 1000$ and compute the average over $T_{\text{SS}} = 100$ Secret Sharer experiments.

The results in Table 3 show the mean rank observed in these experiments. The observed ranks are lower than the value expected in the case of no memorization, i.e. $\frac{N_{\text{SS}}}{2} = 500$. While this observation suggests that some memorization has occurred, the average is still quite close to $500$ signaling that the memorization is either subtle or does not happen often.

To check how often the code and name can be perfectly reconstructed, we also looked at how often a rank of 0 is observed. Note that in a uniform distribution over the rank (meaning no memorization happens), this event should occur $\frac{1}{N_{\text{SS}}} = 0.1\%$ of the time. Table 4 shows that while this event occurs more often than this baseline in our case, the ratio is still low. Improving these metrics and bringing them closer to the no memorization baselines is an important direction for future work.

## 6 FUTURE WORK

**Improving Teaching Process**  Both for sharing examples and sharing instructions, our results show there is room for improvement. Future work could explore other aggregators, ways of introducing learning loops, or other techniques for generating instructions or examples.

**Generalized Settings and Other Modalities**  Future work could also consider more generalized settings, such as cases where teachers are allowed to communicate with each other or are available during inference. Instead of text-based examples and communication, future work can investigate social learning based on other modalities, such as image or audio data. These settings introduce other challenges and require capabilities from the models that are yet to be perfected.

**Alternative Privacy Metrics and Differential Privacy**  While the privacy experiments using Secret Sharer provide some information about privacy in social learning, we do not consider them to be exhaustive. Future work could look into different ways of measuring data leakage in social learning. Furthermore, differential privacy guarantees could be added to social learning by making teachers use recently proposed mechanisms for differentially-private in-context learning [35, 40].

## 7 CONCLUSION

In this work, we introduced the social learning framework which allows language models with access to private data to transfer knowledge through textual communication while maintaining the privacy of that data. In this framework, we identified sharing examples and sharing instructions as basic models and evaluated them on multiple tasks. Furthermore, we adapted the Secret Sharer metric to our framework, proposing a metric for measuring data leakage. The paper evaluates these methods on several datasets, reports results, and outlines directions for future work.

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

## A  ADDITIONAL SIMPLIFYING CONSTRAINTS

We impose the following constraints on the communications between the teachers and the student in our social learning methods:

1. **Teachers do not directly communicate with each other**: teachers are not able to send text messages to each other either directly or via the student. This constraint removes the effect of planning and debate capabilities of the language models.

2. **The query to all teachers is the same**: the student always sends the same message to all the teachers. This constraint removes the need for the student to analyze teacher's knowledge of the task and react based on it.

3. **The conversation flow is fixed**: the tasks requested from the teachers are fixed in advance and do not depend on the conversation. For example, teachers might initially be asked to describe the task and then be prompted with a description from multiple teachers to produce a consolidated version. However, the student will not ask for clarifications about a specific part of the description that is vague. This constraint removes the requirement of models to generate instructions during learning as the prompts can be manually fixed.

To define a social learning method, we have to define the response functions of teachers and the student:

- **Teachers' Response**: We need to define $\mathcal{T}_i(M)$ which is the message sent to the student in response to the message $M$ received from the student. For example if $M$ is a question, $\mathcal{T}_i(M)$ can be the answer based on a teacher's private data.

- **Student's Response**: Since the student sends the same message to all the teachers, we can assume that it replies only after receiving the update from all teachers. The student responds to the message by possibly sending a new message to the teachers and creating an updated prompt $P_{\mathcal{S}}^{\text{New}}$. As such, to define the response function of the student we need to specify $\mathcal{R}_{\mathcal{G}}(M_{\mathcal{T}_1}, M_{\mathcal{T}_2}, \ldots, M_{\mathcal{T}_m}, P_{\mathcal{S}}^{\text{current}})$ as a pair $(P_{\mathcal{S}}^{\text{new}}, M_{\text{next}})$.

The training starts by querying the student to generate the first message to the teachers. Afterwards, the teachers and student alternate responding to each other's messages. Once the training is completed, the final prompt can be used by the student during inference.

## B  DATASETS

In this section, we provide a summary for each of the tasks in our evaluation suite. The exact format used to convert instances of each task to a string given to the language models is provided in Table 4.

**Spam Detection**   We use the SMS Spam dataset [1] which contains a collection of SMS messages classified into spam and not spam classes. We randomly under-sample the dataset (without replacement) to make it balanced. We use a fixed 500 element subset of the under-sampled dataset as the test-set. To convert each example to string we use a basic format which starts with the message's text followed by the class of the message. However, using this format, it is infeasible for the model to perform well when the list of classes are not known. For example, this can happen in the zero-shot or one-shot case where the set of examples contain at most one of the classes. Therefore, we also experiment with another format that provides a list of classes (spam or not spam) before stating the label for the example. The exact format is shown in Table 4. While in the literature normal messages are usually referred to as "ham", we use "not spam" in this work.

**Lambada**   The Lambada dataset [26] is a Cloze task where the last word of a sentence is removed and the task is recovering the word based on the context. In this work, we use the same format used to evaluate GPT-2 [30]. License is CC-BY-4.0.

**Boolean Questions**   BoolQ [8] is a dataset of a context, question, and answer triplets. The model is asked to provide a yes or no answer to the question based on the given context. License is CC-BY-SA-3.0.

**Grade School Math**   We evaluate on the GSM8K dataset [9] which is a set of mathematical questions annotated with the final answer as well as the trace to reach the answer. Solving mathematical

| Dataset | Example Format |
|---------|----------------|
| SMS Spam (Base) | `Text: <Message>`
`Class: <spam/not spam>` |
| SMS Spam (Class List) | `Text: <Message>`
`Class ("spam" or "not spam"): <spam/not spam>` |
| Lambada | `Fill in blank:`

`<Text without last word> ____ -> <last word>` |
| BoolQ | `<Context>`
`Question: <Question>`
`Answer: <Answer>` |
| GSM8K | `Question: <Question>`
`Answer: <Step By Step Reasoning>`
`#### <Final Answer>` |
| Random Insertion | `<Word With Punctuations> = <Original Word>` |

Table 4: Formats used to convert dataset elements to text. The segments enclosed in < and > correspond to placeholders replaced by values from each example.

problems is a known challenging task for language models [31]. Therefore, this task is especially difficult for generating artificial examples since generating a correct example requires solving the task in the process. License is MIT.

**Random Insertion** We also adapt the random insertion artificial dataset from Brown et al. [5]. In this dataset, a random punctuation mark is inserted after each character of a word. The answer to the task is the original word without the punctuation marks. We choose this dataset as the results in Brown et al. [5] show noticeable improvement from having more examples in the few-shot prompt, signaling the importance of having access to good examples or instructions.

## C  SIGNIFICANCE TESTING

We apply a permutation test to understand the significance of our results in comparison to different baselines. In particular, to test the significance of the difference observed in the accuracy of a certain method in comparison to a given baseline, we first combine all the example and output pairs generated by either the baseline or the considered method. We randomly permute the aforementioned pile and break it into a pile with the same number of pairs as the baseline and another pile with the same number of pairs as the considered method. We compute the accuracy of each pile and measure the difference. Repeating this process $10^4$ times allows us to obtain an approximate distribution of the observed difference if the baseline and the considered method's output are not significantly different. We use this distribution to compute the probability of the real difference in accuracy between the baseline and the considered method and report that as the $p$-value. When discussing results, if the $p$-value is below the threshold 0.05 we say the result is significant and state that we could not observe a significant difference otherwise.

## D  DISTANCE OF A GENERATED EXAMPLE TO ITS GENERATION PROMPT

We define a distance metric in order to take into account that the student's prompt can contain multiple examples. In particular, we compute the minimum Levenshtein Distance (with substitution

not allowed) to any substring[1] of the student's prompt. To allow comparability, we normalize this value by the generated example's length and call it the normalized distance. The results are reported in Table 5. The average normalized distance is typically large, indicating that the example is sufficiently different from examples in the prompt. We can also observe that the distance is lower than others in some tasks, namely spam detection and random insertion. We point out that in random insertion almost half the characters are punctuation marks which are limited and can be expected to overlap more often, lowering the distance. Furthermore, the SMS texts are usually short and imitating the format of a spam message can lead to a low distance. That being said, generating novel examples for these tasks may also be more challenging for the model.

| $n$ | Lambada | BoolQ | GSM8K | SMS Spam | SMS Spam (With Class) | Random Insertion |
|---|---|---|---|---|---|---|
| 1 | 0.78 | 0.85 | 0.79 | 0.47 | 0.47 | 0.58 |
| 2 | 0.76 | 0.84 | 0.82 | 0.63 | 0.46 | 0.56 |
| 4 | 0.77 | 0.83 | 0.80 | 0.58 | 0.43 | 0.61 |
| 8 | 0.76 | 0.83 | 0.81 | 0.56 | 0.43 | 0.61 |
| 16 | 0.77 | 0.83 | 0.81 | 0.60 | 0.47 | 0.59 |

Table 5: Average of the normalized distance between each generated example by PaLM 2-S and the prompt used to generate it. Distance is defined as the minimum Levenshtein distance (substitution not allowed) to any substring of the prompt, making the maximum possible distance equal to the generated example's length. Normalization is done by the generated example's length. It can be seen that the average is usually quite high, suggesting that many of the generated examples are significantly different from the real ones provided in the prompt.

## E    MANUAL PROMPTS

The manually written prompts are reported in Table 6.

## F    DETAILED EXPERIMENT RESULTS

The detailed experiment results with standard errors and $p$-values are reported in Table 7.

The results contain cases where the deviation of performance across the runs is quite high, demonstrated by the high reported standard error. We observe that this can happen for multiple reasons. For the spam detection task, this mainly happens when the basic format is used. In this case, the list of classes are unknown to the model and, especially when the number of examples is low, it is possible that the model only receives examples from a single class. We observe that if this class is the spam class, the model uses "ham" to classify non spam messages which is considered the wrong class, thus reducing the accuracy significantly. This is interesting as ham is the terminology typically used in the literature whereas here we use the not spam class. This issue is noticeably improved when the list of classes is provided to the model. High variance is also observed in Boolean Questions. As mentioned earlier, in some runs most generated examples selected by the aggregator were not a yes/no question, which leads to a poor performance. Fortunately, the likelihood for generating such bad examples is low, and such a scenario mainly happens when the number of selected examples $n$ is small. As a result, the high standard error can only be seen for small values of $n$. We can also observe a high standard error in the random insertion task. However, this standard error is also visible in the baseline, suggesting that the model is in general more sensitive to the choice of examples in this task. The root cause of this sensitivity is not clear.

## G    TEACHING A LARGER STUDENT MODEL

In this section, we consider the ability to transfer knowledge to a larger model. This ability is natural to social learning since teachers only share text, enabling knowledge to be transferred between different models of different sizes and architectures. On the other hand, typical gradient-based

---

[1]for a string $s$ with $n$ characters, a substring is defined by a pair $(i, j)$ ($1 \le i \le j \le n$) and refers to the string containing the $i$-th to $j$-th characters of $s$

| Dataset | Manual Instruction |
|---------|--------------------|
| SMS Spam | For the following sms message, determine if it is a spam (e.g. sent by a bot containing advertisement, phishing, spam, etc.) or a real message (sent by a human) by classifying the message into "spam" and "not spam" classes. |
| Lambada | The last word of the last sentence in a passage has been removed. Write the missing word (which is marked by four underscores) after the arrow ->. |
| BoolQ | A passage is given followed by a question. Answer the given question with a simple yes or no based on the given passage. |
| GSM8K | Solve the following math questions. Think step by step and write the steps in your answer. When you are done write the final answer write it (a single number) marked with the prefix #### followed by a space. This answer will be auto-graded so take extra care to follow this format. Do not print anything after the final answer. |
| Random Insertion | A random punctuation mark (or a space) has been inserted after each character of a word. The result is written on the left hand side of the equation below and the right hand side contains the original word. |

Table 6: Manually written instructions used for each task to establish a baseline.

```
Solve the task described below. You may output additional text
however the final answer should be marked with prefix ####
followed by a space.
```

Figure 5: Manually added prefix instruction to specify GSM8K format. No instruction to perform CoT is given.

federated learning methods such as FedAvg [22] and FedOpt [33] require the same model size and architecture to be used everywhere. Given that the largest of language models currently can be only executed on data centers, it would be especially useful to be able to transfer knowledge back to such models.

Table 9 contains the results for teaching a larger student model (PaLM 2-S using smaller teacher models, PaLM 2-XS). As the baseline we compare using original examples either at the student (high bar) or at the teachers (low bar). For all tasks except spam detection we can observe significant

| $n$ | Type | Lambada | BoolQ | GSM8K | SMS Spam | SMS Spam (With Class) | Random Insertion |
|---|---|---|---|---|---|---|---|
| 0 | - | 69.80(0.00) | 68.10(0.00) | 0.00(0.00)[a] | 14.19(0.00) | 92.70(0.00) | 22.00(0.00) |
| 1 | Original | 86.68(0.48) | 89.84(0.10) | 63.59(0.25) | 59.10(7.62) | 94.25(0.27) | 55.56(3.12) |
|   | Generated | 86.65(0.44) | 70.46(7.19) | 63.87(0.76) | 90.22(0.57) | 92.55(0.40) | 53.58(7.89) |
|   | $p$-value | 0.4895 | 0.0000 | 0.3708 | 0.0000 | 0.0023 | 0.0236 |
| 2 | Original | 87.30(0.44) | 90.12(0.03) | 64.20(0.28) | 77.15(9.96) | 94.87(0.25) | 70.04(3.19) |
|   | Generated | 86.70(0.41) | 88.63(0.77) | 63.23(0.60) | 88.17(0.74) | 92.15(0.63) | 65.94(1.75) |
|   | $p$-value | 0.2069 | 0.0000 | 0.1267 | 0.0000 | 0.0000 | 0.0000 |
| 4 | Original | 87.56(0.63) | 90.44(0.07) | 63.59(0.27) | 86.75(8.28) | 95.43(0.53) | 69.74(2.55) |
|   | Generated | 87.98(0.43) | 85.54(3.87) | 63.58(0.48) | 87.77(0.75) | 90.19(0.81) | 69.72(2.42) |
|   | $p$-value | 0.2809 | 0.0000 | 0.5000 | 0.0990 | 0.0000 | 0.5000 |
| 8 | Original | 88.36(0.54) | 90.53(0.07) | 64.05(0.23) | 96.02(0.27) | 96.75(0.11) | 74.50(1.15) |
|   | Generated | 88.05(0.27) | 88.73(0.88) | 63.38(0.47) | 86.45(0.88) | 91.51(0.97) | 69.22(3.42) |
|   | $p$-value | 0.3246 | 0.0000 | 0.2164 | 0.0000 | 0.0000 | 0.0000 |
| 16 | Original | 88.40(0.67) | 90.42(0.08) | 63.55(0.28) | 96.48(0.17) | 97.02(0.07) | 73.52(1.11) |
|   | Generated | 89.04(0.23) | 89.94(0.08) | 63.71(0.35) | 87.98(1.18) | 91.08(1.57) | 72.36(1.01) |
|   | $p$-value | 0.1747 | 0.0756 | 0.4266 | 0.0000 | 0.0000 | 0.1023 |

[a]GSM8K uses a special format to mark the answer. The model inevitably always fails when no instruction or examples are provided to it to clarify this special format. Adding the prefix stated in Figure 5 to clarify the format yields accuracy 16.38%.

Table 7: Accuracies and $p$-values reported in Table 1 with more precision. Standard error of the mean is reported in parentheses.

| Method | Lambada | BoolQ | GSM8K | SMS Spam | SMS Spam (With Class) | Random Insertion |
|---|---|---|---|---|---|---|
| Zero-Shot | 69.80 | 68.10 | 0.00 | 14.19 | 92.70 | 22.00 |
| Manual | 77.45 | 90.18 | 15.62 | 93.95 | 94.22 | 34.9 |
| 8-shot Original Examples | 88.36(0.54) | 90.53(0.07) | 64.05(0.23) | 96.02(0.27) | 96.75(0.11) | 74.50(1.15) |
| 8-shot Artificial Examples | 88.05(0.27) | 88.73(0.88) | 63.38(0.47) | 86.45(0.88) | 91.51(0.97) | 69.22(3.42) |
| GPT3.5 Generated Inst. | 82.81(1.87) | 90.12(0.07) | 4.11(2.27) | 85.38(8.70) | 95.38(0.37) | 59.22(4.76) |
| PaLM 2-S Generated Inst. | 85.12(0.91) | 88.74(1.36) | 0.00(0.00) | 92.90(0.04) | 93.44(0.39) | 40.38(9.88) |

Table 8: Accuracies and $p$-values reported in Table 2 with more precision. Standard error of the mean is reported in parentheses.

improvement over using the original examples from the small model. The gap is especially large for smaller values of $n$ (e.g. 1-shot) where an improvement can be observed on all tasks. While this improvement is expected given the larger size of the student's model, it highlights the success of generated examples to transfer the knowledge and demonstrates the benefit of having such mechanism. For larger values of $n$, the small model already performs quite well on the spam detection task and as a result, no significant improvement from the knowledge transfer can be observed in these cases. Noticeably, in most cases for Lambada and GSM8K no significant difference could be observed between using the artificially generated examples and using private examples directly at the student.

We discussed the challenges encountered when generating new examples for the spam detection and Boolean Question tasks in Section F. We observe that when using a smaller model, the same challenges persist and are sometimes exacerbated. As a result, the generated examples can sometimes perform poorly as can be observed for 1-shot inference in the Boolean Questions task and 2-shot inference for the spam detection task without list of classes. In these cases, a high standard error is typically observed as the model only sometimes fails to generate good examples.

# H  VOTING AGGREGATOR

In this section we explore using a more sophisticated aggregator than the random aggregator and assess its effect on performance. In particular, we consider an aggregator that adheres to the following voting process:

| $n$ | Type | Student | Lambada | BoolQ | GSM8K | SMSSpam | SMS Spam (With Class) | Random Insertion |
|---|---|---|---|---|---|---|---|---|
| 1 | Original | PaLM 2-XS | 74.6 | 81.1 | 9.3 | 61.8 | 54.3 | 11.8 |
| | Original | PaLM 2-S | **86.7** | 89.8 | 63.6 | 59.1 | 94.3 | 55.6 |
| | Generated PaLM2-XS | PaLM 2-S | **86.7** | 72.2* | 57.2* | 75.9* | 92.4* | 50.9* |
| 2 | Original | PaLM 2-XS | 73.7 | 80.9 | 16.0 | 72.2 | 75.1 | 19.9 |
| | Original | PaLM 2-S | **87.3** | **90.1** | **64.2** | 77.2 | 94.9 | 70.0 |
| | Generated PaLM2-XS | PaLM 2-S | **87.8** | **89.8** | **63.6** | 59.7* | 87.9* | 66.1* |
| 4 | Original | PaLM 2-XS | 81.5 | 81.1 | 19.2 | 90.4 | 94.9 | 25.1 |
| | Original | PaLM 2-S | **87.6** | 90.4 | **63.6** | 86.8 | 95.4 | 69.7 |
| | Generated PaLM2-XS | PaLM 2-S | **88.1** | 82.8* | **63.9** | 94.1* | 93.8* | 51.5* |
| 8 | Original | PaLM 2-XS | 86.2 | 81.9 | 18.7 | 95.7 | 96.5 | 31.4 |
| | Original | PaLM 2-S | **88.4** | **90.5** | 64.1 | 96.0 | **96.8** | 74.5 |
| | Generated PaLM2-XS | PaLM 2-S | **89.1** | **90.2** | 63.6* | 94.6* | **96.1** | 63.3* |
| 16 | Original | PaLM 2-XS | 87.3 | 82.6 | 17.7 | 96.3 | 96.2 | 30.2 |
| | Original | PaLM 2-S | **88.4** | 90.4 | **63.6** | 96.5 | 97.0 | 73.5 |
| | Generated PaLM2-XS | PaLM 2-S | **89.2** | 89.1* | **63.8** | 94.6* | 94.0* | 61.8* |

Table 9: Performance of teaching a larger student model. The performance of an PaLM 2-XS student using original examples is reported as the low bar baseline whereas the performance using original examples and PaLM 2-S student constitutes the high bar baseline. A star marks statistically significant results from the high bar baseline. We bold cells where no statistically significant difference was detected to emphasize that in many cases the examples generated using social learning perform as well as the original ones. The average accuracy across 5 runs is reported. Table 10 reports the same values with more precision.

1. Before beginning the generation process, the aggregator asks each teacher to create a evaluation dataset by holding out a subset of its data, not used for generating the artificial examples.

2. After each generation, as specified in Section 2.2.2, the aggregator is queried with a set of artificially generated candidates. As a response, the aggregator sends the list of all candidates to all teachers asking them to select the best candidate.

3. Each teacher computes the likelihood of each candidate separately as a continuation of its held-out evaluation dataset normalized by the length of that candidate and votes for the candidate that scores the highest. The teachers' votes are sent back to the aggregator.

4. The aggregator selects the candidate with the most votes.

As before, the process of generating candidates, voting and selecting the highest voting candidate is repeated until the desired number of examples is generated to be included in the student's prompt. We call this aggregator the voting aggregator.

We compare the performance of using the voting aggregator against using the random aggregator in Table 11. We observe that the benefit of using the voting aggregator varies depending on $n$. For very small values of $n$ (e.g. $n = 1$) the performance is even worse than using the random aggregator for some tasks. Though the observed difference is not always significant, this may suggest that the top-voted example, though possibly better formatted, might not be sufficient to fully describe the task as a single example which encourages looking for better aggregation mechanisms. At the other end of the spectrum, we observe no significant difference for very high values of $n$, e.g. $n = 16$. We hypothesize that in this case given the large number of examples, these examples contain most of the information even when they are selected randomly. However, for middle range values of $n$ where the choice of the examples is important and there is some freedom in using different combinations, we observe a more pronounced difference when using a voting aggregator. In this case, for most of the tasks an improvement is observed in the accuracy (though not always significant) when using the voting aggregator. The exception is the spam detection task where using the voting aggregator tends to hurt the performance regardless of the magnitude of $n$. We noticed that this is because when using

| $n$ | Type | Student | Lambada | BoolQ | GSM8K | SMS Spam | SMS Spam (With Class) | Random Insertion |
|---|---|---|---|---|---|---|---|---|
| 1 | Original | PaLM 2-XS | 74.61(2.15) | 81.05(1.02) | 9.28(2.04) | 61.75(4.55) | 54.30(1.88) | 11.80(3.37) |
| | Original | PaLM 2-S | 86.68(0.48) | 89.84(0.10) | 63.59(0.25) | 59.10(7.62) | 94.25(0.27) | 55.56(3.12) |
| | Generated PaLM2-XS | PaLM 2-S | 86.72(0.72) | 72.17(6.46) | 57.21(4.83) | 75.94(7.89) | 92.42(1.51) | 50.92(3.25) |
| | $p$-value | | 0.4883 | 0.0000 | 0.0000 | 0.0000 | 0.0012 | 0.0000 |
| 2 | Original | PaLM 2-XS | 73.65(3.22) | 80.94(0.98) | 16.03(0.26) | 72.15(8.16) | 75.11(9.25) | 19.94(2.60) |
| | Original | PaLM 2-S | 87.30(0.44) | 90.12(0.03) | 64.20(0.28) | 77.15(9.96) | 94.87(0.25) | 70.04(3.19) |
| | Generated PaLM2-XS | PaLM 2-S | 87.84(0.53) | 89.75(0.15) | 63.55(0.58) | 59.73(6.78) | 87.88(3.55) | 66.10(1.25) |
| | $p$-value | | 0.2240 | 0.1346 | 0.2211 | 0.0000 | 0.0000 | 0.0000 |
| 4 | Original | PaLM 2-XS | 81.48(2.90) | 81.11(0.22) | 19.15(0.65) | 90.43(1.41) | 94.92(0.66) | 25.14(1.94) |
| | Original | PaLM 2-S | 87.56(0.63) | 90.44(0.07) | 63.59(0.27) | 86.75(8.28) | 95.43(0.53) | 69.74(2.55) |
| | Generated PaLM2-XS | PaLM 2-S | 88.05(0.26) | 82.82(4.48) | 63.85(0.64) | 94.09(0.44) | 93.87(0.51) | 51.46(0.43) |
| | $p$-value | | 0.2372 | 0.0000 | 0.3846 | 0.0000 | 0.0019 | 0.0000 |
| 8 | Original | PaLM 2-XS | 86.20(0.43) | 81.90(0.27) | 18.61(0.44) | 95.73(0.59) | 96.45(0.23) | 31.44(2.43) |
| | Original | PaLM 2-S | 88.36(0.54) | 90.53(0.07) | 64.05(0.23) | 96.02(0.27) | 96.75(0.11) | 74.50(1.15) |
| | Generated PaLM2-XS | PaLM 2-S | 89.12(0.11) | 90.18(0.08) | 62.64(0.77) | 94.62(0.29) | 96.13(0.30) | 63.26(2.46) |
| | $p$-value | | 0.1263 | 0.1511 | 0.0497 | 0.0029 | 0.0844 | 0.0000 |
| 16 | Original | PaLM 2-XS | 87.26(0.24) | 82.62(0.35) | 17.68(0.45) | 96.34(0.35) | 96.16(0.39) | 30.22(1.70) |
| | Original | PaLM 2-S | 88.40(0.67) | 90.42(0.08) | 63.55(0.28) | 96.48(0.17) | 97.02(0.07) | 73.52(1.11) |
| | Generated PaLM2-XS | PaLM 2-S | 89.18(0.33) | 89.08(0.99) | 63.75(0.71) | 94.62(0.47) | 93.95(1.11) | 61.76(3.49) |
| | $p$-value | | 0.1196 | 0.0000 | 0.4120 | 0.0002 | 0.0000 | 0.0000 |

Table 10: Accuracies and $p$-values reported in Table 9 with more precision. Standard error of the mean is reported in parentheses.

voting, the bias of the model toward one class as discussed in the previous section becomes amplified. Our results suggest that additional research is required to find better aggregators that can improve the performance further which we leave as an area for future work.

| $n$ | Method | Lambada | BoolQ | GSM8K | SMSSpam | SMS Spam (With Class) | Random Insertion |
|---|---|---|---|---|---|---|---|
| 1 | Random | 86.7 | 70.5 | **63.9** | **90.2** | 92.6 | 53.9 |
| | Voting | 86.5 | **86.6***  | 60.5* | 87.3* | 93.2 | **56.6*** |
| 2 | Random | 86.7 | **88.6** | 63.2 | **88.2** | **92.2** | 65.9 |
| | Voting | **87.9*** | 85.8* | **64.8*** | 83.7* | 88.1* | **67.8*** |
| 4 | Random | 88.0 | 85.5 | 63.6 | **87.8** | 90.2 | 69.7 |
| | Voting | 88.2 | **89.8*** | 64.8 | 84.1* | 91.1 | **72.9*** |
| 8 | Random | 88.1 | 88.7 | 63.4 | **86.5** | **91.5** | 69.2 |
| | Voting | 88.2 | **89.5*** | 64.0 | 84.8* | 89.5* | **71.4*** |
| 16 | Random | 89.0 | 89.9 | 63.7 | 88.0 | **91.1** | 72.4 |
| | Voting | 88.8 | 89.7 | 63.5 | 87.9 | 89.4* | 72.8 |

Table 11: Comparison of the performance of PaLM 2-S when using the voting and random aggregators. A star marks statistically significant results from the random to the voting aggregator according to the permutation test. We bold the cells that are better and have statistical significance. The change in performance when using the voting aggregator seems to depend on the value of $n$. While for the large values of $n$ the results do not change and the random aggregator performs better for the small values of $n$, middle values of $n$ benefit from using the voting aggregator. The exception is the spam detection task where using a voting aggregator always reduces performance, possibly due to the bias of model towards not spam messages. The average accuracy across 5 runs is reported. Table 12 reports the same values with more precision.

| $n$ | Method | Lambada | BoolQ | GSM8K | SMS Spam | SMS Spam (With Class) | Random Insertion |
|---|---|---|---|---|---|---|---|
| 1 | Random | 86.65(0.44) | 70.46(7.19) | 63.87(0.76) | 90.22(0.57) | 92.55(0.40) | 53.58(7.89) |
| | Voting | 86.50(0.28) | 86.63(2.60) | 60.52(1.07) | 87.31(0.61) | 93.15(0.24) | 56.56(3.66) |
| | $p$-value | 0.3122 | 0.0000 | 0.0000 | 0.0001 | 0.1711 | 0.0019 |
| 2 | Random | 86.70(0.41) | 88.63(0.77) | 63.23(0.60) | 88.17(0.74) | 92.15(0.63) | 65.94(1.75) |
| | Voting | 87.87(0.28) | 85.80(2.81) | 64.75(0.33) | 83.74(2.19) | 88.06(1.71) | 67.84(1.29) |
| | $p$-value | 0.0001 | 0.0000 | 0.0367 | 0.0000 | 0.0000 | 0.0228 |
| 4 | Random | 87.98(0.43) | 85.54(3.87) | 63.58(0.48) | 87.77(0.75) | 90.19(0.81) | 69.72(2.42) |
| | Voting | 88.18(0.33) | 89.76(0.22) | 64.82(0.42) | 84.11(1.94) | 91.05(0.96) | 72.90(2.07) |
| | $p$-value | 0.2408 | 0.0000 | 0.0691 | 0.0000 | 0.1127 | 0.0001 |
| 8 | Random | 88.05(0.27) | 88.73(0.88) | 63.38(0.47) | 86.45(0.88) | 91.51(0.97) | 69.22(3.42) |
| | Voting | 88.22(0.20) | 89.53(0.30) | 64.03(0.32) | 84.78(1.96) | 89.52(1.12) | 71.36(0.98) |
| | $p$-value | 0.2809 | 0.0092 | 0.2216 | 0.0230 | 0.0019 | 0.0098 |
| 16 | Random | 89.04(0.23) | 89.94(0.08) | 63.71(0.35) | 87.98(1.18) | 91.08(1.57) | 72.36(1.01) |
| | Voting | 88.76(0.25) | 89.65(0.09) | 63.53(0.36) | 87.85(3.04) | 89.44(1.26) | 72.82(2.18) |
| | $p$-value | 0.1613 | 0.1956 | 0.4195 | 0.4427 | 0.0089 | 0.3116 |

Table 12: Accuracies and $p$-values reported in Table 11 with more precision. Standard error of the mean is reported in parentheses.

In the case of the spam dataset, the random aggregator does better than the voting aggregator. This is because language models are inherently biased towards non-spam examples, meaning the voting process leads to a class imbalance.

The choice of the aggregator is thus highly dataset-dependent. When the quality of candidate examples and instructions is high, random selection does well. When it is not, it becomes more important to pick generated examples well.

## I  CANARY DESIGNS

Table 13 shows the canaries we use for Secret Sharer experiments.

## J  EXAMPLES AND INSTRUCTIONS OUR METHODS GENERATE

Tables 14, 15, 16 show examples and instructions as generated by our methods.

| Dataset | Codes | Names |
|---------|-------|-------|
| LAMBADA | The secret ____ is 1234 -> code | `<name>` was on the way to buy bread at the bakery. On the way to the bakery he came across a dog. Instead of continuing to the ____ `<name>` followed the dog. -> bakery |
| GSM8K | The first secret code is 1234. The second secret code is 5678. What is the product of the secret codes?

The product is 1234 * 5678 = <<<1234 * 5678>>> #### 7006652 | `<name>` went to the bakery to buy two dozen cookies. `<name>` then shared the cookies equally with five friends. How many cookies did each person get?

There are 24 cookies and 6 people, so each person got 24 / 6 = <<24 / 6>> cookies. #### 4 |

Table 13: The canary patterns used to generate the canaries for each dataset.

| Dataset | Generated Example |
|---------|-------------------|
| Lambada | ```Fill in blank:```

```The car went into the water. The water came inside```
```the car. The water went up to the driver's ___ ->```
```seat``` |
| Lambada | ```Fill in blank:```

```The last time he'd eaten a mango was on the way to```
```Sri Lanka,where he and his wife were honeymooning.```
```As he took a bite,he closed his eyes and smiled at```
```the sweet taste.He'd never tasted anything so ___ ->```
``` delicious``` |
| BoolQ | ```Question: is the super bowl the biggest event of the```
``` football season?```
```Answer: yes``` |
| BoolQ | ```Question: do sunflowers track the sun?```
```Answer: yes``` |
| GSM8k | ```Question: Nancy is returning her overdue books to```
```the library. She owes $0.50 cents each on 8 books,```
```plus a flat $2.00 fee for having at least one book```
```that's over a week overdue. How much does she have```
```to pay total```
```Answer: First multiply the late fee per book by the```
```number of books: $0.50/book * 8 books = $```
```<<0.5*8=4.00>>4.00. Then add the "more than a week```
```overdue" fee to the answer from the previous step to```
``` find the total: $4.00 + $2.00 = $<<4+2=6.00>>```
```#### 6``` |
| GSM8k | ```Question: An 800 gram cake is cut into 8 equal```
```pieces. If 6 of the pieces are sold, what is the```
```total weight of the cake that was sold?```
```Answer: The cake has 8 pieces and weighs 800 grams,```
```so each piece weighs 800 grams / 8 pieces =```
```<<800/8=100>>100 grams. If 6 of the pieces are sold,```
``` the total weight is 6 pieces * 100 grams / piece =```
```<<6*100=600>>600 grams```
```#### 600``` |
| SMS Spam | ```Text: New year's wish: I wish that we all could have```
``` this same type of conversation all of the time.```
```Class: not spam``` |
| SMS Spam | ```Text: Hi James, just wanted to check you're ok. Speak```
``` later```
```Class: not spam``` |
| Random Insertion | ```d`a)i\$l*y = daily``` |
| Random Insertion | ```c`o)m\$p*u.t/e`r computer``` |

Table 14: A selection of cases where the teacher generates useful examples. In all of these cases, the generated examples follow the correct format and look like original training examples.

| Dataset | Generated Example |
|---------|-------------------|
| Lambada | Fill in blank:

After a month of waiting, the couple was finally ready to start a family. They had been married for ten years, and now it was time for them to ___ -> start a family |
| Lambada | Fill in blank:

The crowd of people outside of the stadium grew and grew. Many wore ___ -> the team's colors |
| BoolQ | Question: where is john lennon from?
Answer: England |
| GSM8k | Question: A square pyramid with a square base has a square side 2 feet long. How many feet do the pyramids side total to the top,
Answer: Each face of the pyramid has a side length of 2 feet, so the height is 2, and there is a total of 4 sides. The total height then is 4x(2+2)= <<4x(2+2)=12>>12 feet.
#### 12 feet. |

Table 15: A selection of cases where the teacher does not generate useful examples. The Lambada examples are incorrect because multiple words are filled in the blank. The BoolQ example is not actually a binary question. And the GSM8k example contains a math mistake.

| Dataset | Generated Instruction |
|---|---|
| Lambada | The task is to fill in the blank with a word that makes sense in the context. The blank is indicated by ———. 

 The input is a piece of text with a blank. The output is a word that makes sense in the context. |
| BoolQ | The task is to fill in the blank with a word that makes sense in the context. The blank is indicated by ———. 

 The input is a piece of text with a blank. The output is a word that makes sense in the context. |
| GSM8k | The input is a math problem. The output is a detailed solution to the problem, with all intermediate steps and calculations shown. The output should be in the same format as the input, with all special markings and formatting preserved. |
| SMS Spam | You are given a text message. You are expected to classify it into two classes: spam and not spam. Spam messages are usually advertisements or promotional messages. Not spam messages are usually personal messages. 

 The output should be a string "spam" or "not spam". |
| Random Insertion | The input is a string with some special characters. The special characters are `, $, *, ., and #. The special characters are used to indicate the position of the spaces in the original string. The output is the original string with the spaces removed. |

Table 16: A selection of instructions generated by the teacher.

