# OpenReview forum: "Social Learning: Towards Collaborative Learning with Large Language Models"
_ICLR.cc/2025/Conference — Submitted to ICLR 2025_

### Official Review · Reviewer_dUDz · 2024-10-26

**Soundness:** 1
**Presentation:** 3
**Contribution:** 3
**Rating:** 3
**Confidence:** 3

**Summary:**

The manuscript proposes a "social learning" framework for privacy-preserving LLM knowledge transfer. The method proceeds by using "teacher" models with access to private training data that output some knowledge representation. This knowledge representation is aggregated by a "student" and passed to an LLM in tandem with a test-time query. No private data ever enters the student model. The approach is validated on a variety of question-answering/classification benchmarks including GSM8K and BoolQ using the PaLM 2 model family.

**Strengths:**

* The core idea is quite clever, and the research question is well-scoped — I would summarize the core research question as follows: "Since LLMs are so good at following instructions, why not use LLMs to generate better instructions from data for privacy preservation?"
* The writing is intuitive. It's easy to follow the paper's main line of argument.

**Weaknesses:**

* [W1] Lack of formalization: While math is certainly not a prerequisite to a successful machine learning paper, without an underlying formalization of what the task is, it's hard to understand the proposed approach. Notation in 2.1 are limited to defining variables describing the teachers, student, and datasets, but not how they interact. To be fair, I find the examples and descriptions are informative and useful. However, they are a little bit too hand-wavey/open to interpretation due to imprecise language such as "agents collaborate" and "augment [the student's] knowledge." Other places of confusion include "[the student] must select a subset of the information [from teachers]," but subsequently, in 2.2.1, the teachers provide natural-language instructions, making me confused whether "subset" is used in the colloquial sense or the set-theoretic sense.

While such language is great for building intuition, I do not believe it can form the *sole* basis of a problem formulation in a technical paper. Furthermore, the lack of such formalisms makes it much harder to draw connections to related works in machine learning including and beyond LLMs.

* [W2] Unclear connection to social learning theory: I found the analogy to social learning theory distracting since the connection to the proposed approach seems to be "agents can learn tasks by examples/demonstrations," which sounds like supervised learning in general. Without clear reasoning for this connection, I think the reference actually obfuscates the proposed approach; I.e., if the reference was completely removed, one could still imagine a convincing paper that describes how natural language instructions from multiple agents can be aggregated to design better prompts for a "student LLM."
* [W3] Weak baseline: While results for a "voting" aggregator are in an Appendix, the main results use a random aggregation strategy. This means that, effectively, for each test-time example, the student is only using information from *one* teacher at a time. Thus, the experiments do not actually validate whether leveraging *multiple* teachers is helpful. See "Questions" for a suggested stronger baseline.
* [W4]: Missing ablations/sensitivity analysis: The number of teachers feels like a core design choice — any insights in how varying the number of teachers could improve the student's performance? Are more teachers always helpful, or do we see an U-shaped curve (i.e., too many "teachers" pulls the student in "different directions")?
* [W5] References to the model/knowledge distillation and curriculum learning literature are missing. These references could help contrast the proposed method, which aims to summarize "knowledge" in the abstract from teacher models, to previous distillation approaches, that aim to distill *parameters,* and curriculum learning approaches, which explicitly *select* data subsets for the student. The previous works seem to adopt a strictly narrower sense of "knowledge" (model distillation) or the mechanism of transfer (curriculum learning) the fact that LLMs can follow natural-language instructions provides new avenues to distill models, which this paper exemplifies. This paper is not the first to use the "student-teacher" verbiage in the context of machine learning. Some examples for consideration (in case the authors would like to expand their related works):
- [Hinton et al., '14](https://arxiv.org/abs/1503.02531), NeurIPS workshop, and [Ba & Caruana '14](https://arxiv.org/pdf/1312.6184), NeurIPS main conference — earliest usage of the "student teacher" verbiage I could find in the context of knowledge transfer. Similar ideas presented in [Vapnik & Ismailov '15](https://www.jmlr.org/papers/volume16/vapnik15b/vapnik15b.pdf?ref=https://githubhelp.com), JMLR.
- [Bucilu et al., '06](https://www.cs.cornell.edu/~caruana/compression.kdd06.pdf), KDD — earliest instance of "transfer knowledge from larger -> smaller model" that I am aware of.
- [Bengio et al., '09](https://dl.acm.org/doi/10.1145/1553374.1553380), ICML — original "curriculum learning" paper.

* [W6 — very, very minor] I find it unconventional to bold cells highlighting no statistical significance, since it's such a common practice in ML papers to bold the best-performing approach. But I was able to understand Table 1 fine.

**Questions:**

* **Re [W1]:** Could the authors propose a formalization of the problem setup (i.e., a "student" is a mapping from .... A "teacher" is a mapping .... The training algorithm can be described as ... At inference time, given some user query, the student returns [object])? Here's a starting point that I wrote based on my understanding of the paper (please tweak/correct as needed, or propose a completely different one):

For simplicity, we consider a supervised binary classification task, with datasets of the form $ (x_i, y_i)_{i=1}^{N} $, where $x_i \in \mathcal{X}, y_i \in \\{0, 1\\}$.

A **teacher** $\\mathcal{T_j}$ is an arbitrary "agent" ($j = 1, \dots, m$) that maps private datasets $\\mathcal{D}\_{j}$ to "summaries" $\mathcal{I}_j$ (*e.g.,* parameters of a trained ML model, a natural language instruction).

A **student** is a tuple $ \\mathcal{S} = (\\mathcal{S}\_{agg}, \\mathcal{S}\_{LLM} ) $, where $\\mathcal{S}_{agg}: \mathcal{I}\_1 \times \dots \times \mathcal{I}\_m \mapsto s\_{inst}$ aggregates summaries from teachers into an "instruction" $s\_{inst}$ and $  \\mathcal{S}\_{LLM}: s\_{inst}, q(x) \mapsto \hat{y}$ concatenates the instruction to a user query $q(x)$ for inference-time data point $x$ and outputs a prediction $\hat{y}$.

Then:
* What is the student's "objective?" What are the students' inputs/outputs at training vs. inference time?
* How does the student "learn" to solve underlying tasks (i.e., what objective would a "good" student minimize)? Note that this does not need to be a formal "loss function" that can be numerically optimized (the proposed method doesn't do so, and that's OK) — I'm asking for a way to characterize the "quality" of a student in the abstract.
* Under the formulation, what is the output space of the teacher (e.g., natural language instructions? model parameters/vectors? synthetic examples $\in \mathcal{X}$?), and what are some choices of "aggregators?"

For me, [W1] is a "dealbreaker." I am very happy to re-assess the quality of the paper and approach if I see a formulation provided in the rebuttal for consideration that answers the above sub-questions, with a promise to incorporate into Section 2.1, but I do not think I can justify changing my score without seeing such a formulation first.

**Questions I am curious about, but need not be answered immediately:**
* Re W4: In Section 4.1, "The dataset is distributed between teachers randomly so all teachers will have the same data distribution." Wouldn't violating this assumption potentially *improve* performance; i.e., each teacher is largely "redundant" with respect to the others currently? E.g., re W3, my intuition is that under more diverse teachers, you'd see a larger gap between the random vs. voting aggregator (i.e., Appendix H).
* Re W3: What are other possible aggregators to consider? For example, one could implement a "summarization" aggregator that combines natural language summaries outputted by each teacher via an LLM.
* I'm confused about the utility of the memorization experiment. E.g., in the spam classification example in the intro, suppose my student is constrained to only ever output "yes" or "no," and suppose that the teachers output natural-language instructions that have verifiably low overlap with their training data. Wouldn't P(student outputs canary phrase) = 0 be guaranteed? In other words, isn't there already some inherent privacy-preservation? Can the authors speak to a concrete example where a student could produce a harmful output with respect to privacy preservation?

---

### Official Review · Reviewer_EkNm · 2024-11-01

**Soundness:** 2
**Presentation:** 2
**Contribution:** 2
**Rating:** 3
**Confidence:** 4

**Summary:**

This paper proposes to consider a new framework of social learning among LLM agents. It allows agents with access to private data to transfer knowledge through textual communication while maintaining data privacy. It differs from conventional LLM agent settings in terms of the presence of private data. The paper proposes to use either generated instruction or generated examples to transfer knowledge. Experiments show that collaboration could lead to higher performance as compared to single-model zero-shot performance.

**Strengths:**

1. The setting of social learning with consideration of dataset privacy is valuable to consider.
2. The way to perform majority voting for example selection is interesting.

**Weaknesses:**

1. There is a lack of either privacy guarantee or empirical verification. Section 5 on memorization could not prove reduced private data leakage as claimed.
2. There is a lack of discussion and experimentation in settings where data distributions are different across.
3. The experiments section is not very well organized and explained.

**Questions:**

1. For both methods, i.e., sharing instructions and sharing examples, they rely on the quality of the LLM-generated content. How to ensure the generated context does not compromise the privacy of the teacher’s private dataset? Prompting it does not guarantee such privacy.
2. The proposed social learning protocol mainly aims to address knowledge transfer with privacy protection. However, the main problem of the paper is the lack of either privacy guarantee or empirical verification. The section “Memorization” fails to do so. Comparing the reconstruction likelihood in the in-context learning pipeline only accesses how memorization happens in in-context learning, but does not give any idea of whether the generated dataset does not contain sensitive information from the original dataset. Hence, it is misleading to claim that the proposed method helps to reduce private data leakage using these experiments.
3. In line 159, “Some of these continuations might be discarded due to concerns such as privacy or faulty generation”. How is that done? Is it by manual checking or heuristics?
4. It is not reasonable to assume the data distributions among teacher models are the same. There is a lack of discussion and experimentation on settings where data distributions are different across, with possibly different teacher models, architecture and hence abilities. Do you have results on this?
5. The teacher has no knowledge about the ability of the student model while generating synthetic examples for them to “learn”. This might pose potential issues as the teacher does not understand the student. In extreme cases, for example, consider the student has a short context window and then a teacher that always generates long examples will fail (the teacher does not know that the student cannot take long examples). Maybe having such discussions could be helpful.

---

### Official Review · Reviewer_GQJJ · 2024-11-01

**Soundness:** 3
**Presentation:** 2
**Contribution:** 2
**Rating:** 5
**Confidence:** 3

**Summary:**

This paper

**Strengths:**

Originality: The paper proposes an interesting and novel technique for learning from private data inspired by the concept of social learning. It makes intuitive sense that, rather than sharing data or gradient updates, an LLM could be perfectly capable of instead distilling what it learns from its observations into advice for another LLM.
Significance: The paper proposes a technique for sharing lessons from data without exposing the underlying data, which relates to broader concerns around privacy.

**Weaknesses:**

Quality:
a) While the canary method is a standard way of measuring memorization, I have concerns about its ecological validity in this setting as a measure of privacy preservation. To use the private spam/no spam SMS message example from the paper, one way that the teacher might naturally communicate part of a spam filter could be to say that a particular person (let's call them Alice) is never spam. However, if the user does not want the fact that they text Alice to be revealed, then the message "Alice is never spam" would leak private information. In contrast, the teacher in the paper would never have any particular incentive to communicate a canary message to the student, since random password strings would not help on the task. As such, the canary method does not seem to particularly support the idea that this method is necessarily private. It would be helpful to see if privacy is preserved in a setting where performance could be improved by sharing private information.
Clarity:
b) I think the paper probably could explain the experimental setup and results sooner, and not relegated so many experimental details to the appendices.
c) The experimental setup could be clarified a bit more succinctly. What is the teacher given (8 random examples?)? What does the student see?

**Questions:**

1) What exactly does the student observe?
2) It seems like the experimental setup is mostly about the teacher sharing formatting with the student, rather than something qualitatively more similar to "information". Does that seem correct? Have the authors conducted experiments where the teacher sees (for example) the chain of thought for GSM8k and needs to communicate a hint?

---

### Official Review · Reviewer_JfoZ · 2024-11-04

**Soundness:** 3
**Presentation:** 2
**Contribution:** 2
**Rating:** 5
**Confidence:** 4

**Summary:**

This paper proposed a new way to transfer knowledge in a collaborative learning setting where multiple teachers want to collaborate to teach a good student. A new metric is been proposed to measure how much privacy information in leaked via this collaborative learning process. Experimental results have shown that this new collaborative learning is effective.

**Strengths:**

1. The metric proposed by this paper to measure information leakage is novel to me.
2. The experimental results show that the two collaborative learning approaches proposed in this paper is effective in transferring knowledge.

**Weaknesses:**

1. The way to share knowledge either via instruction sharing or synthetic exemplars is not new. Existing works have already used synthetic samples generated by the teacher models to teach the student model.

2. The new metric for information leakage is not compared with other existing work like membership inference attacks [1]. What's the advantages of this new metric compared to that?

3. This paper focus on collaborative learning setting, however, no results showing that this collaborative learning scheme is effective or not. One way to show this is that multiple teacher models will outperform a single teacher model, which is not shown in this paper. Another important aspect is aggregation method which is also not shown in instruction sharing of the method. Can the author provide results on this?


[1] Shokri, R., Stronati, M., Song, C., & Shmatikov, V. (2017, May). Membership inference attacks against machine learning models. In 2017 IEEE symposium on security and privacy (SP) (pp. 3-18). IEEE.

**Questions:**

see weakness

---

### Meta-Review · Area_Chair_3TjX · 2024-12-17

**Metareview:**

The paper proposes a social learning method for LLMs, which aims to achieve information sharing among LLMs.

The reviewers think that the idea is novel and clever and hence has good potential.

However, the reviewers also identified some notable weaknesses: insufficient discussions w.r.t. previous works, insufficient experimental results showing the effectiveness of the proposed method, lack of clarity when explaining the experimental setting, lack of mathematical formalization, missing important ablation etc.

Therefore, the paper has room for improvement and hence rejection is recommended.

**Additional Comments On Reviewer Discussion:**

The authors did not submit rebuttal.

---

### Decision · Program_Chairs · 2025-01-22

Reject